# Time of Onset and Risk Factors of Renal Involvement in Children with Henoch-Schönlein Purpura: Retrospective Study

**DOI:** 10.3390/children9091394

**Published:** 2022-09-15

**Authors:** Nicolina Stefania Carucci, Giulia La Barbera, Licia Peruzzi, Antonella La Mazza, Lorena Silipigni, Angela Alibrandi, Domenico Santoro, Roberto Chimenz, Giovanni Conti

**Affiliations:** 1Pediatric Nephrology and Rheumatology Unit, AOU G Martino, University of Messina, 98125 Messina, Italy; 2Pediatric Nephrology Unit, Regina Margherita Department, Azienda Ospedaliero-Universitaria Città Della Salute E Della Scienza, 10126 Torino, Italy; 3Unit of Statistical and Mathematical Sciences, Department of Economics, University of Messina, 98125 Messina, Italy; 4Unit of Nephrology and Dialysis, Department of Clinical and Experimental Medicine, University of Messina, 98125 Messina, Italy

**Keywords:** Henoch-Schönlein purpura, kidney, children, nephritis, risk factors, onset, abdominal pain, laboratory data

## Abstract

Background: Henoch-Schönlein purpura (HSP) is a common systemic vasculitis in children, involving the skin, musculoskeletal system, gastrointestinal tract and kidneys. Some studies in children have shown possible risk factors linked with the development and severity of HSP Nephritis (HSPN). The aim of this study was to research predicting factors for the development of HSPN. Methods: We retrospectively evaluated 132 pediatric patients with HSP, according to EULAR/PRINTO/PRESS criteria. All patients were screened for HSPN by urinalysis. Finally, we compared demographic, clinical and laboratory data in HSP patients with and without nephritis. Results: The median age at HSP diagnosis [6.2 (2.6–17.5) vs. 5.5 (0.8–15.4) years, *p* = 0.03] and the incidence of abdominal pain (48 vs. 27%, *p* = 0.01) were significantly higher in HSPN patients. No differences were evidenced regarding gender, allergic diseases, skin recurrences, gastrointestinal involvement, musculoskeletal involvement, scrotal involvement, and laboratory data (white blood cell count, neutrophil count, lymphocyte count, platelet count, C-reactive protein, erythrocyte sedimentation rate, and blood concentration of IgA). Conclusions: The age at diagnosis and abdominal pain were independent risk factors for renal involvement in HSP patients. However, due to the retrospective nature of this study, further long-term and prospective studies will be necessary.

## 1. Introduction

Henoch-Schönlein purpura (HSP), nowadays called immunoglobulin A (IgA) vasculitis, is the most common systemic vasculitis in pediatric age, with an annual incidence of 10–20 per 100,000 [1,2,3,4,5]. More than 90% of patients are younger than 10 years old and most HSP patients’ age is between 4 and 6 years [3,6]. The major pathogenetic hypothesis of HSP is that polymeric IgA1 are deposited in the skin, gastrointestinal tract capillaries and glomerulus of HSP patients. In detail, galactose-deficient IgA1 dominant immune deposits were suggested to play a major role in HSP Nephritis (HSPN) [4,7].

In 2010, the European League Against Rheumatism (EULAR), the Paediatric Rheumatology International Trials Organisation (PRINTO) and the Paediatric Rheumatology European Society (PRES) proposed new classification criteria for pediatric HPS patients [1]. The classification criteria defined HSP in the presence of non-thrombocytopenic palpable purpura or petechiae (mandatory criterion) with lower limb predominance, plus one of four criteria: abdominal pain; histopathology (IgA deposition in tissue biopsy); arthritis or arthralgia; and renal involvement [8]. HSP often has a self-limiting course and the extrarenal symptoms usually resolve rapidly without complications; however, 16.3% of patients particularly recur with skin and gastrointestinal involvement [4].

The long-term outcome depends primarily on the extent of renal involvement, termed HSP nephritis (HSPN) [3,9]. The clinical spectrum of HSPN ranges from the relatively common transitory microscopic hematuria and/or low-grade proteinuria to nephritic or nephrotic syndrome, rapidly progressive glomerulonephritis or even renal failure [1,2,3]. The incidence of HSPN ranges from 20 to 50% and the evolution to end-stage renal disease is reported to be 2–5% [10]. Currently, the International Study of Kidney Disease in Children (ISKDC) classification is the one most used for histologic analysis of HSPN lesions. Recently, the Oxford classification is used more for IgA nephropathy, which has many histological features similar to HSPN. There are multicenter studies in progress to evaluate the application of the Oxford classification in HSPN [2].

The risk factors associated with HSPN are not well known, although some epidemiologic and clinical features have been suggested to have a predictive role [3]. Possible biochemical markers gained from routinely performed laboratory tests are rarely investigated in patients with systemic vasculitis to practically predict and prevent the morbidities and complications [4]. Blood neutrophil-to-lymphocyte ratio (NLR) was investigated in many diseases as a marker of inflammation. NLR and platelet-to-lymphocyte ratio (PLR) are hypothesized to be potentially useful markers of developing renal involvement in early phase of disease, but few studies have reported this association [5]. In this retrospective study, the epidemiological, clinical and laboratory characteristics of our HSP patients were analyzed to determine the initial risk factors for HSPN development. We also aimed to study the relationship between NLR and PLR in renal involvement in HSP.

## 2. Materials and Methods

We enrolled 132 pediatric patients with HSP followed at the Pediatric Nephrology and Rheumatology Unit of our University Hospital between May 2003 and June 2020. All the patients fulfilled validated EULAR/PRINTO/PRES criteria for HSP vasculitis [8]. Patients with less than 1 year of follow-up were excluded. Demographic data (gender and age at diagnosis), clinical data (allergic diseases, skin recurrences, gastrointestinal involvement, abdominal pain musculoskeletal involvement, renal involvement, and scrotal involvement), and laboratory findings (white blood cell (WBC) count, neutrophil count, lymphocyte count, platelet count, C-reactive protein (CRP), erythrocyte sedimentation rate (ESR), and blood level of IgA) were retrospectively retrieved from the medical records of our hospital. NLR was calculated as the ratio of neutrophil count to lymphocyte count, and PLR as the ratio of platelet count to lymphocyte count.

The parents of the patients have given consent to the execution of laboratory tests and the collection of clinical data. Allergic diseases included allergic conjunctivitis, allergic rhinitis, asthma and atopic dermatitis. Skin recurrence was defined as new cutaneous lesions after at least one asymptomatic month after the initial HSP episode. Arthralgia was characterized by joint pain without joint swelling or limitation of motion. Arthritis was defined by joint swelling or pain with limitation of motion. Gastrointestinal (GI) involvement was defined as abdominal pain within 14 days of diagnosis of palpable purpura, and/or occult or macroscopic GI tract bleeding, and/or bowel intussusception on ultrasound examination. Scrotal involvement could include scrotal pain with redness and swelling and/or an imaging abnormality (e.g., epididymitis-orchitis, epididymitis). Normal ranges for the age and gender of the child were used to interpret the laboratory test results [11].

All patients were screened by urinalysis every 7 days in the first 30 days of diagnosis, every 14 days for the next 60 days and later every month for the first year. HSP patients were divided into two groups: with and without HSPN. Renal involvement was defined by the presence of macrohematuria, microhematuria (urine sediment showing >5 red blood cells per high-power field), and/or proteinuria (24-h urinary protein >4 mg/m^2^/h (or >5 mg/kg/day) or a proteinuria/creatininuria ratio > 0.2 (mg/mg)). Nephrotic syndrome was defined by the presence of heavy proteinuria (≥50 mg/kg/day (or ≥40 mg/m^2^/h), or a proteinuria/creatininuria ratio >2 mg/mg), hypoalbuminemia (less than 2.5 g/L), and edema. Acute nephritic syndrome was defined as hematuria, non-nephrotic proteinuria, elevated blood pressure, decreased urine output, and edema. Minimum criteria for acute kidney injury included an increase in serum creatinine to ≥1.5 times baseline or urine volume <0.5 mL/kg/h for 6 h.

Renal biopsy was performed when patients showed nephritic or nephrotic syndrome, acute kidney disease, and/or persistent proteinuria (>3 months). Kidney biopsy was graded according to histological findings proposed by the ISKDC, preferred by us to the Oxford classification, though not yet widely used in HSPN [2,12].

### Statistical Analysis

Statistical analyses were performed using the Statistical Package for Social Sciences (SPSS) version 22.0 for Windows. Results were presented as median (range) or mean ± standard deviation (SD) for continuous variables and number (%) for categorical variables. A Kolmogorov–Smirnov test was used to evaluate normal distribution of data. A Mann–Whitney test was used to compare continuous variables. A Chi-squared test was used to compare categorical variables. In addition, receiver operating characteristic (ROC) curve analysis was performed to determine the optimal cutoff value for age at diagnosis. Multivariable logistic regression analysis was used to determine possible independent predictors of HSPN; the dependent variable was the presence of HSPN and the independent variables were those statistically significant in univariate analysis. Results were presented as odds ratios with 95% confidence intervals. For all statistical tests, a *p* value less than 0.05 was considered of statistical significance.

## 3. Results

### Demographic, Clinical and Laboratorial Data and Treatments

A total of 132 patients with HSP were identified. The median age at diagnosis was 5.7 (0.8–17.5) years and 69 (52%) of them were boys. The median follow-up duration was 1220 (425–5300) days. All patients involved in our study had palpable purpura at the time of diagnosis. Thirty-one (23%) patients had history of at least one allergic disease. Fifty-eight (64%) patients had arthritis or arthralgia. GI involvement was present in 68 (52%) patients, 49 (72%) of whom complained of abdominal pain; five (7%) patients presented with intussusception. Sixty-one (46%) patients had suffered from at least one skin recurrence during follow-up. Eighty-five (64%) patients needed a treatment; of them, 61 (71%) patients were treated with oral prednisolone, 17 (20%) patients with high dose intravenous pulse methylprednisolone, and 7 (8%) patients received combined corticosteroid and cytotoxic agent treatment (cyclophosphamide, mycophenolate mofetil).

Some laboratory tests performed at the onset of HSP were also analyzed, which included WBC count, neutrophil count, lymphocyte count, NLR, platelet count, PLR, CRP, ESR, and IgA. These laboratory tests were performed at the onset of HSP in 96 patients. Leukocytosis was detected in 14/96 (15%) patients, an increased neutrophil count in 22/96 (23%) and decreased lymphocyte count in 14/96 (15%), related with median range for patient’s age. Platelet count was increased in 17/96 (18%) patients; ESR and CRP were increased in 58/96 (60%) and in 69/96 (72%), respectively. An increased level of serum IgA was detected in 26/96 (27%) patients.

HSPN was diagnosed in 65/132 (49%) patients. Of them, 26/65 (40%) HSPN patients had renal involvement at admission, 38/65 (58%) HSPN patients developed renal involvement in the first 10 days, 47/65 (72%) HSPN patients in the first month, 59/65 (91%) HSPN patients in the first 6 months. Six/96 (9%) HSPN patients developed renal involvement after 6 months. One patient presented with hematuria and proteinuria 37 days before the onset of purpura. Isolated hematuria was observed in 32%, isolated proteinuria in 9%, and both (hematuria and proteinuria) in 52%; nephrotic syndrome was identified in 2 patients, acute kidney injury associated with nephritic/nephrotic syndrome in 1 patient, and acute kidney injury associated with nephritic syndrome in 1 patient. None of the patients developed end-stage renal disease or were submitted to renal replacement therapy. Percutaneous renal biopsy was performed in 8/65 (12%) HSPN patients. The ISKDC histological findings showed grade II in 4/8 (50%), grade III in 3/8 (38%), and grade IV in 1/8 (12%).

Demographic characteristics and clinical data were compared between 132 HSP patients who developed nephritis (HSPN) and those that did not (No HSPN) (Table 1). The median age at HSP diagnosis [6.2 (2.6–17.5) vs. 5.5 (0.8–15.4) years, *p* = 0.03] and the incidence of abdominal pain (48 vs. 27%, *p* = 0.01) were significantly higher in HSPN patients compared to those without renal complication. No differences were evidenced regarding gender, allergic diseases, skin recurrences, gastrointestinal involvement, musculoskeletal involvement, or scrotal involvement (Table 1).

Laboratory data performed at the onset of HSP were present in 96 patients. No significant differences were evidenced regarding WBC count, neutrophil count, lymphocyte count, NLR, platelet count, PLR, CRP, ESR, and IgA in HSPN patients compared to those without renal involvement (Table 2).

Both univariate and multivariate analysis confirmed that age at diagnosis (OR = 1.14; 95% CI 1.02–1.29; *p* = 0.03) and abdominal pain (OR = 2.53; 95% CI 1.20–5.32; *p* = 0.02) were independent risk factors for renal involvement with statistical significance (Table 3).

## 4. Discussion

In this retrospective analysis of 132 HSP patients, we attempted to determine the effect of some possible clinical and laboratory risk factors of renal involvement in childhood HSP.

The incidence of HSPN varies greatly depending on ethnic and geographic background. Renal involvement in children with HSP has been reported to occur in 26.2% [4] to 58% [1]. This variability is linked to epidemiologic characteristics of the cohort, level of assistance (primary or tertiary center), type of study (retrospective or prospective), criteria of HSPN and duration of follow up. Consistent with the literature, the present study observed HSPN in almost one half (49%) of patients [1,4].

At disease onset, nephritis occurred from 20 to 80% of HSP patients, as observed herein (40%) [1]. According to previous studies, in 75–100% of patients, renal involvement develops within 4–6 weeks of the onset of the rash, and in almost all patients within the first 90 days [5]. In our patients, renal involvement was detected within 30 days from disease onset in 72% of patients, and within 6 months in 91%. However, in 6 patients (9%), renal disease was diagnosed later. Some authors suggest monitoring HSP patients for 6 months for the reason that HSPN is very rare after 6 months in their cohorts [2,13]. Furthermore, the SHARE initiative suggests monitoring HSP patients with blood pressure measurements and urinalysis for at least 6–12 months, even if the initial data are normal [14]. In light of our findings, we recommend close monitoring for up to 1 year in order to detect late onset of renal involvement.

The majority (85%) of our HSP patients presented hematuria with or without proteinuria and 9% presented isolated proteinuria, whereas severe renal abnormalities were rarely observed, as also previously reported [15]. The prognosis was favorable in our patients with absence of progression to end-stage renal disease, but the relatively good outcomes may be due to shorter follow-up times compared to other studies.

Renal biopsy is considered the gold standard to establish severity of HSPN according to ISKDC classification [16]. According to previous studies [1], percutaneous renal biopsy was performed in 8/65 (12%) HSPN patients selected for nephritic syndrome, nephrotic range proteinuria, acute kidney disease, and persistent proteinuria (>3 months). Male-to female ratio was 1.7. The ISKDC histological findings showed grade II in 4/8 (50%), grade III in 3/8 (38%), and grade IV in 1/8 (12%).

Increasing attention has recently been paid to the risk factors of renal damage of HSP because their early identification has important implications for treatment and follow-up [3,6]. In our study, we included several epidemiological features, clinical manifestations, and laboratory findings. Univariate and multivariate analysis revealed that age at diagnosis (OR = 1.14; 95% CI: 1.02–1.29; *p* = 0.03) and abdominal pain (OR = 2.53; 95% CI: 1.20–5.32; *p* = 0.02) are associated with an increased risk of nephritis (Table 3). The majority of HSP patients are diagnosed before the age of 10 years with a peak between 4 and 6 years [9,17]. The present study demonstrated a median age at diagnosis of 5.7 (0.8–17.5) years. Different authors concluded that an age older than 4 years [18], 6 years [17], 7 years [19], 8 years [20] and 10 years [3] was an independent risk factor for renal involvement. Our study confirmed that older children were at higher risk of kidney disease than younger children, although we could not determine a cutoff value.

Chan et al. showed that boys are at higher risk for kidney disease in HSP than girls [3]. Conversely, some authors reported that HSPN is more common in girls [21,22]. Our study indicated that sex had no impact on renal damage onset in HSP patients. However, we found that the incidence of HSP was slightly higher in boys than in girls, but boys had a lower incidence (44.6%) than girls in HSPN cohort, indicating that girls are at greater risk of renal damage.

Supported by the literature, we studied the impact of allergic diseases on HSPN. Wei et al. showed that atopic dermatitis was a risk factor for HSP and HSPN in children [9]. On the contrary, Chen et al. suggests that atopic children were more likely to develop HSP, but allergic diseases were not associated with an increase in renal involvement [10]. Likewise, we analyzed the effect of a large number of common allergic diseases, but we found no correlation with HSPN onset.

Gastrointestinal and musculoskeletal symptoms are the predominant manifestations of HSP. Chan et al. showed that digestive tract symptoms were strongly related to renal involvement [3]. We found that gastrointestinal symptoms had no influence on the onset of renal damage of HSP, although abdominal pain was significantly higher in patients with nephritis (Table 2).

We observed skin recurrences in 40% of patients. Most studies identify an important association between persistent purpura or recurrence and HSPN [23]. Our analysis did not confirm this result.

In order to predict the risk of HSPN, several inexpensive and practical biomarkers were investigated [1,2,3,4,5,6,9,10,17,19,22]. A meta-analysis published by Mao et al. suggested that hemoglobin was a risk factor for kidney damage in HSP patients [6]. In the same year, Chan et al. showed that elevated leukocyte and platelet counts predict the onset of renal involvement [3]. In contrast, we found no relevance between these laboratory parameters and nephritis. The NLR has been widely used to define the severity of inflammation [5]. Because neutrophils are the main effector cells in HSP, it was suggested to investigate NLR as a potential risk factor for HSPN. Ekinci et al. indeed demonstrated that neutrophil count and NLR were higher in patients with biopsy-proven nephritis than without [4]. Contrary to these results, in our cohort, neutrophil count and NLR did not differ between patients in regard to the presence of HSPN. From the reviewed literature, some patients with HSPN had elevated ESR, CRP, IgA [24,25]. Conversely, according to our analysis, these parameters are not related to HSPN.

One of the strengths of this study was the inclusion of a quite large number of HSP patients followed in a Italian tertiary center that fulfilled the validated EULAR/PRINTO/PRES criteria [8]. The long-term follow-up was relevant. The main limitation of our study was the retrospective design, with potential missing data and recall bias. Moreover, the patients of our center mainly come from the south of Italy, therefore the genetic background and environmental factors may be different from other cohorts, resulting in ethnicity selection bias.

## 5. Conclusions

In conclusion, our investigation suggests that age at diagnosis and abdominal pain are associated with HSPN. However, due to the retrospective nature of this study, further long-term and prospective studies are needed.

## Figures and Tables

**Table 1 children-09-01394-t001:** Comparison of demographic characteristics and clinical data between 132 patients with and without HSPN.

Characteristics at Onset (*n* 132)	HSPN (*n* 65)	No HSPN (*n* 67)	*p*-Value
**Demographic data**			
**Age, *years, m* ^1^**	**6.2 (2.6–17.5)**	**5.5 (0.8–15.4)**	**0.03**
Sex, *n* (%)			
Female	36 (55%)	27 (40%)	
Male	29 (45%)	40 (60%)	0.08
**Clinical data**			
Allergy, *n* (%)	15 (23%)	16 (24%)	0.91
Skin recurrence, *n* (%)	34 (52%)	27 (40%)	0.17
Gastrointestinal involvement, *n* (%)	37 (57%)	31 (46%)	0.22
**Abdominal pain, *n* (%)**	**31 (48%)**	**18 (27%)**	**0.01**
Arthritis/arthralgia, *n* (%)	39 (60%)	46 (69%)	0.30
Other systemic involvement, *n* (%)	8 (12%)	11 (16%)	0.50

^1^*m*, median range.

**Table 2 children-09-01394-t002:** Comparison of laboratory data between 96 patients with and without HSPN.

Laboratory Data (*n* 96)	HSPN (*n* 47)	No HSPN (*n* 49)	*p*-Value
↑ WBC ^1^, *n* (%)	8 (18%)	6 (12%)	0.41
↑ N ^2^, *n* (%)	12 (27%)	10 (20%)	0.41
↓ L ^3^, *n* (%)	6 (14%)	8 (16%)	0.78
NLR ^4^, *n* (*m* ^5^)	2.0 (0.5–9.5)	1.9 (0.6–9.6)	0.84
↑ PLT ^6^, *n* (%)	7 (16%)	10 (20%)	0.64
PLR ^7^, *n* (*m*)	125.7 (52.5–469.5)	122.2 (50–376.5)	0.976
↑ CRP ^8^, *n* (%)	27 (59%)	31 (60%)	0.93
↑ ESR ^9^, *n* (%)	29 (61%)	40 (81%)	0.08
↑ IgA, *n* (%)	12 (25%)	14 (28%)	0.75

^1^ WBC, White Blood Cell count; ^2^ N, Neutrophil count; ^3^ L, Lymphocyte count; ^4^ NLR, Blood neutrophil-to-lymphocyte ratio; ^5^
*m*, median range ^6^ PLT, Platelet count; ^7^ PLR, platelet-to-lymphocyte ratio; ^8^ CRP, C-reactive protein; ^9^ ESR, erythrocyte sedimentation rate.

**Table 3 children-09-01394-t003:** Results of univariate and multivariate analysis.

	Univariate Analysis			Multivariate Analysis		
Variable	OR	CI 95%	*p*-Value	OR	CI 95%	*p*-Value
Age at onset	1.14	1.02–1.28	**0.03**	1.14	1.02–1.29	**0.03**
Sex	0.54	0.27–1.09	0.08			
Allergy	0.96	0.43–2.14	0.91			
Skin recurrence	1.63	0.82–3.24	0.17			
GI ^1^ involvement	1.54	0.77–3.05	0.22			
Abdominal pain	2.48	1.20–5.14	**0.01**	2.53	1.20–5.32	**0.02**
Arthritis/arthralgia	0.69	0.34–1.40	0.30			
Other systemic involvement	0.72	0.27–1.91	0.50			
↑ WBC ^2^	1.62	0.52–5.09	0.41			
↑ N ^3^	1.49	0.57–3.88	0.41			
↓ L ^4^	0.85	0.27–2.67	0.78			
NLR ^5^	0.93	0.77–1.13	0.45			
↑ PLT ^6^	0.78	0.27–2.25	0.64			
PLR ^7^	1.00	0.99–1.00	0.70			
↑ CRP ^8^	0.96	0.42–2.20	0.93			
↑ ESR ^9^	0.37	0.12–1.14	0.09			
↑ IgA	0.85	0.30–2.36	0.75			

^1^ GI, gastrointestinal; ^2^ WBC, White Blood Cell count; ^3^ N, Neutrophil count; ^4^ L, Lymphocyte count; ^5^ NLR, Blood neutrophil-to-lymphocyte ratio; ^6^ PLT, Platelet count; ^7^ PLR, platelet-to-lymphocyte ratio; ^8^ CRP, C-reactive protein; ^9^ ESR, erythrocyte sedimentation rate.

## Data Availability

Not applicable.

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
