# Peer review of "Time of Onset and Risk Factors of Renal Involvement in Children with Henoch-Schönlein Purpura: Retrospective Study"

_children, 2022, doi:10.3390/children9091394_

Round 1
Reviewer 1 Report
HSP, a common systemic vasculitis in children, affects renal tissue in various proportions, only a few patients reaching the stage when renal replacement therapy is needed.
The authors describe older age at presentation and abdominal pain as risk factors to the development of nephritis in these patients with HSP.
Major comments:
Even though it was a single center retrospective study, there is no mention of it being approved/exempt from review by the local Institutional Review Board.
The screening of patients are reported similar to following a protocol, as expected for a prospective study, with very consistent reporting of symptoms and data. There is no mention of missing data.
Authors used ISKDC pathology classification of renal lesions, but a report from Pediatric Rheumatology 2019, 17:10 concludes that none of the current classifications have been validated, though the Oxford classification appears to be used in practice more often, despite the fact that it was initially used in IgA nephropathy. Authors should explain why ISKDC classification was used by them instead of the Oxford classification.
Introduction – line 63 – the reference [2] is about Oxford classification, though the authors state the reference is about ISKDC classification
Minor comments:
Several syntax and spelling errors throughout the manuscript are noted, and I just mention a few:
Abstract – line 23 – “predictors factors” should be “predicting factors” or just “predictors”
Abstract – line 32 – “blood level” should be “blood concentration”
Introduction – line 42 – “patients aged” should be “patients’ age is”
Introduction – line 47 – rephrase the first sentence, as it is confusing
Tables – values should be expressed as “0.xx” not “0,xx”
Discussion – line 236 – “sex have” – use the appropriate verb declination
Discussion – line 243 – suggest to rephrase “did not associate with”
Discussion – line 252 – second sentence – reformulate the beginning – suggest – “Most studies identified…”
Author Response
We thank the reviewer for the comment and we hope to respond satisfactorily to improve the article Major Comments- The parents of the children enrolled retrospectively in the study had signed an informed consent to perform the tests.
- Some laboratory data are missing. In fact we have the laboratory data of 96 of 132 patients enrolled in the study. The laboratory data evaluated are common blood tests: CBC, CRP, ESR and IgA
- We have considered the ISKDC pathology classification of renal lesions, because the Oxford classification currently has less use in HSP nephritis. The Oxford classification is currently validated only in IgA nephropathy. There are some multicenter studies in progress, in which we are also participating, to evaluate whether it is possible to apply the same criteria of the Oxford classification used in IgA nephropathy also in HSP nephritis. We have considered your suggestion and therefore we explain this choice in the article and we have also changed reference 2
Reviewer 2 Report
This is a well-conducted and well-presented study on a topic that clinicians are still looking for definitive data to guide clinical practice. As noted by the authors, the strength is a large cohort of patients in a single center with a protocolized practice pattern and long follow up. These results confirm 2 risk factors that have been found in other studies, but also notably do not confirm several other symptom and laboratory-based factors that other studies have found to correlate with development of HSPN. If these results are confirmed with prospective, multi-center studies it would streamline much clinical decision-making. The authors rightly note that the retrospective and single-center nature of the study to limit the ability to extrapolate the findings to current clinical practice.
I only suggest correction of a few spelling and grammar mistakes, but the data and the presentation are quite good.
Line 152 - change to "one patient presented with hematuria..."
Line 196 - change "pf" to "of"
Line 215 - change "established" to "establish"
Line 236 - change "sex have" to "sex has"
Line 254 - change "didn't" to "did not"
Line 259 - change "in contrary" to "in contrast"
Author Response
We thank the Reviewer for having enjoyed our article We have correct the spelling and grammar mistakesRound 2
Reviewer 1 Report
Appreciate changes made as suggested. Line 69 - spell check Oxford
Author Response
Thanks to the reviewer for appreciating the changes. We corrected the typo